# On Lower Bounds for the Number of Queries in Clustering Algorithms

## Abstract

We consider clustering with the help of an oracle when all queries are made at once and the clusters are determined after all the responses are received. We determine the minimum number of queries required to completely cluster all items. We also consider active clustering with the help of an oracle when a number of queries are made in the first round and the remaining queries are made in the second round based upon the responses to the queries from the first round. We determine a lower bound for the number of queries required to completely cluster all items based upon an analysis of the number of queries made in the first round. Finally, for the two round case, we give results which characterize the problem of clustering in the second round based upon the number of queries made in the first round.

## 1 Introduction

We consider the process of classification of a set of items into clusters by an oracle. This classification is useful, for example, when sets of images such as images of dogs are to be grouped into sets of similar images, such as breeds of dogs. The algorithms we consider identify the clusters by making queries about pairs of items that provide information as to whether the two items are in the same cluster or not. In the real world, this type of classification is presently performed by platforms such as Amazon Mechanical Turk, Zooinverse, Planet Hunters, etc. Korlakai Vinayak & Hassibi (2016); Yi et al. (2012). A clustering algorithm may be designed for sending one query at a time and making the next query based upon responses to the queries already sent out. That method of making queries is called active querying. Active querying may not be a time efficient way of determining the clusters because of the time involved in waiting for the answers to come. If a large number of queries are required, it may not be practical to make queries in that manner.

Another method of making queries is called passive querying wherein all the queries are sent out at once and the clustering is determined when the answers are returned. This approach is not as computationally efficient as active querying because of the potential redundancy of queries involved when the queries are made at once and nothing is known about the clusters, especially when there are many items to be classified.

Herein, we examine a third approach which combines elements of active and passive querying which involves sending out a number of queries at the same time and deciding the next step based upon an analysis of all the responses received to those queries. We call the process of sending out the queries and analyzing the responses a "round." We call the set of queries in a round a "batch." See Guo & Schuurmans (2007); Gissin & Shalev-Shwartz (2019) for more on querying in batches.

We determine the minimum number of queries required to complete clustering in one round and two round algorithms and provide some additional analysis of the two round case based upon the number of queries made in the first round.

## 2 Problem Setup

Formally, the problem we consider is the following. Consider a set $S$ of $n$ unique items. Without loss of generality, we can identify the items with the integers $1, \ldots, n$. The set is partitioned into $k$ disjoint subsets,

which we call clusters $C_1, \ldots, C_k$. Without loss of generality, we can assume that the sizes of the clusters satisfy $|C_1| \leq |C_2| \leq \ldots \leq |C_k|$. We define $l_i = |C_i|$, $i = 1, \ldots, k$.

The objective of an active clustering algorithm is to identify the clusters by making queries about pairs of items that provide information as to whether the two items are in the same cluster or not. That is, we study an active clustering mechanism with pairwise similarity queries. We represent a query by an ordered pair of integers corresponding to the natural numbers assigned to the items. If a query involves items $a$ and $b$, then the query is denoted by $(a, b)$, $a < b$, $a \in 1, \ldots, n$, $j \in 1, \ldots, n$. The space of all queries is $A = \{x = (a, b) : a \in 1, \ldots, n, b \in 1, \ldots, n, a < b\}$. The set $A$ consists of $\binom{n}{2} = \frac{n \times (n-1)}{2}$ elements.

For query $x = (a, b) \in A$, we can define a function $f$ that is equal to 1 if the items $a$ and $b$ belong to the same cluster and 0 if $a$ and $b$ belong to different clusters. The algorithm places the items in the same cluster if $f(x)$ takes the value 1. One can think of $f$ as an indicator function that indicates whether two elements of a query are in the same cluster.

$$f(x) = \begin{cases} 1, & \text{if } a \in C_i, \ b \in C_i \text{ for some } i = 1, \ldots, \text{k} \\ 0, & \text{if } a \in C_i, \ b \in C_j, \text{ for some } i \neq j. \end{cases} \tag{1}$$

We assume that the responses to the queries are not subject to error, which is sometimes expressed by saying that the workers are perfect. The algorithms that we consider in this paper produce the clusters by processing the queries and grouping items into "preclusters" based upon the responses to the queries that are available until all clusters have been determined. A precluster is a group of items that has been determined to belong to the same cluster at some stage of the process of determining the clusters. It is important to note that the preclusters may not be complete clusters. For query $x = (a, b)$, if $a$ and $b$ are found to be in the same cluster, then we place the two items in the same precluster. An item that has not been found to be in the same precluster as any other item is a precluster of size 1.

A key observation we use in our proofs is that the number of preclusters decreases by at most 1 with each query that is processed. We envision the process of determining the clusters as starting with a collection of $n$ one-element preclusters, corresponding to the $n$ items of the set. Every time a response to a query indicates that the elements of the query that are in different preclusters are in the same cluster, i.e., $f(x) = 1$, a new precluster is created by combining the two preclusters containing the elements of the query into one precluster. For example, if we start with $n$ preclusters, the first time we come across a query $x$ for which $f(x) = 1$, we combine the two one-element clusters that contain the elements of the query into a new precluster, so we now have $n - 1$ preclusters.

An active querying algorithm may be designed for sending one query at a time and making the next query based upon responses to the queries already sent out. That method of making queries is called active querying. Active querying may not be an efficient way of determining the clusters because of the time involved in waiting for the answers to come. If a large number of queries is required, it may not be practical to make queries in that manner. To remedy the lack of time effectiveness of active querying, another approach is to send out a number of queries at the same time and decide the next step based upon an analysis of all the responses received to those queries. We call the process of sending out the queries and analyzing the responses a "round". We call the set of queries in a round a "batch." A one-round algorithm is an algorithm designed to complete the determination of the clusters in one round. That is, we need to send out enough queries so that the responses will be sufficient to determine all the clusters. In this paper, we consider the question of how many queries are required by algorithms with varying numbers of rounds.

## 3 Related Work

The work of Mazumdar & Saha (2017b) provides information-theoretic lower and upper bounds on the number of queries needed to cluster a set of $n$ iterms into $k$ clusters. They note that $O(nk)$ is an upper bound on the query complexity and that $O(nk)$ is also a lower bound for randomized algorithms (Davidson et al., 2015). They obtain asymptotic upper and lower bounds on the number of queries required to recover the clusters when side information is provided in the form of certain similarity values between each pair

of elements. The work of Mazumdar & Saha (2017a) provides information-theoretic lower bounds for the number of queries for clustering with noisy queries, which can be made interactively (adaptive queries) or up-front (non-adaptive). Of particular interest in the context of the present work is their result for the case when the number of clusters, $k$, is $k \geq 3$, stating that if the minimum cluster size is $r$, then any deterministic algorithm must make $\Omega\left(\frac{n^2}{r}\right)$ queries even when query answers are not subject to error, to recover the clusters exactly. (Mazumdar & Saha, 2017a) mentions that this shows that adaptive algorithms are much more powerful than their nonadaptive counterparts, but that comment does not take into account the fact that adaptive algorithms in practice may require much more time to run, which could make them impractical. The previous asymptotic result is consistent with Theorem 1 of the present work but our result provides a tighter lower bound and is not asymptotic.

We remark that a framework for active clustering by an oracle has been considered since at least 2004 (Basu et al., 2004), when a method for selecting queries was provided with all the queries made at once. The numbers of queries required to completely cluster the items were not provided. The authors of Balcan & Blum (2008) analyze an adaptive querying setting where the clustering is not unique and needs only satisfy several relations with respect to the data. Additionally, in each iteration, the worker is presented with a set of queries and the worker is given a proposed clustering wherein the worker selects a cluster in the proposed clustering to split into two clusters or selects two clusters to merge. Information theoretic bounds on the number of queries are provided. The authors of Awasthi et al. (2017) extend the work of Balcan & Blum (2008) with a similar problem formulation which examines the scenario of clustering iteratively beginning with any initial clustering and split/merge requests of the clusters at each iteration. They note that this form of clustering is easy for untrained workers since the split/merge requests only require a high-level understanding of the clusters. The work of Korlakai Vinayak & Hassibi (2016) demonstrates lower bounds for adaptive querying algorithms. The work considers the setting of a unique underlying clustering and uses Hoeffding's inequality and a version of the law of iterative logarithm to get bounds on probabilities of successfully recovering clusters. Another line of work that is similar to active clustering is explored by Vempaty et al. (2014), which looks at $M$-ary classification wherein queries are made to classify an object into one of $M$ fine-grained categories. These categories are used to determine which of the clusters an object belongs in.

Other works also analyze similar settings. The work of Ashtiani et al. (2016) proposes a framework for the popular $k$-means clustering where the algorithm utilizes active clustering to speed up $k$-means clustering where a dissimilarity function of the items is assumed and can be used to perform $k$-means clustering. In their work, the results of a set of answers to pairwise similarity queries are used in conjunction with a $k$-means clustering algorithm to cluster the items in polynomial time. They provide bounds on the number of queries that are required to achieve polynomial time $k$-means clustering. The work of Ailon et al. (2017) provides similar results to Ashtiani et al. (2016) for approximate clustering without the additional assumptions upon the structure of the clusters when responses to a number of queries are given. The work of Chien et al. (2018) develops approximate center-based algorithms with size information provided by same-cluster queries, but with constraints upon the size of the smallest cluster and they introduced outliers into the analysis using different methods of proof. In their work the query complexity is reduced both in the case of noisy and noiseless responses. Their problem setting is related to the work in Ashtiani et al. (2016) through the use of query models for improving clustering and the lower bound on the number of queries is provided asymptotically as a function of the number of queries.

## 4 Results

**Observation 1.** *Consider a one-round algorithm. We first observe that in general, if nothing is known about the number of clusters or their sizes, then in order to ensure that all the clusters are determined, $\binom{n}{2}$ queries are needed.*

For example, suppose that there are clusters, that is, each item is in its own cluster. In that case, if a single query $x = (a_1, a_2)$ is omitted, no algorithm would be able to detect that items $a_1$ and $a_2$ are in the same cluster without additional information about the number and sizes of clusters. If at least two items ($a_1$ and

$a_2$) are in single-item clusters, then if the query $x = (a_1, a_2)$ is omitted, there would be no way to determine whether $i_1$ and $i_2$ are in the same cluster.

**Observation 2.** *We obtain a different lower bound for the number of queries if the number of clusters, $k$, is known. In this case, at least one query can always be removed, and the clustering can still be completed. This is because if the number of preclusters has been reduced to $k + 1$ and one query is remaining, then we do not need to make that query to complete the determination of clusters because we can deduce that the response to the last query must be 1. When the number of clusters, $k$, is known, the lower bound is $\binom{n}{2} - 1$.*

However, if two queries are removed, then in general no algorithm may be able to determine all clusters even if $k$ is known. For example, if two queries are removed and there are two single-element clusters, say, $\{a_1\}$ and $\{a_2\}$, and one two-element cluster, say $\{a_3, a_4\}$, then after all $\binom{n}{2} - 2$ queries have been analyzed we may still be left with $k + 1$ preclusters. This is because if the two queries that are removed are $(a_1, a_2)$ and $(a_3, a_4)$, then $a_1, a_2, a_3$, and $a_4$ would remain as one-element preclusters because there would be no way of determining if $a_1$ and $a_2$ are in the same cluster or if $a_3$ and $a_4$ are in the same cluster. In other words, at least one of the queries, $(a_1, a_2)$ or $(a_3, a_4)$, are necessary to complete the determination of the clusters and reduce the number of preclusters to $k$. The previous example suggests that the number of queries required to determine all the clusters depends upon the size of the two smallest clusters. For a given active querying algorithm, let $B \subset A$ be the set of all queries that are utilized by the algorithm. We seek bounds upon the size $|B|$ of the set $B$.

Our main result for one-round algorithms is the following:

**Theorem 1.** *Let $l_1$ and $l_2$ be the sizes of the smallest cluster and the second smallest cluster, respectively, and $C_1$ and $C_2$ be the smallest cluster and the second smallest cluster, respectively. Any active clustering algorithm with perfect workers that determines all the clusters with one round requires more than $\frac{(n - (l_1 + l_2)) \times n}{2}$ queries.*

*Proof.* Noting that $B$ is the set of all queries that are utilized by the algorithm, let $S_i = \{j \in 1, \ldots, n : (i, j) \in B \text{ or } (j, i) \in B\}$, where $i \in 1, \ldots, n$. In other words, $S_i$ is the set of items for which the algorithm utilizes queries involving item $i$. If the algorithm utilizes, for each item $i$, at least $p$ queries involving item $i$, i.e., $|S_i| \geq p$, $i = 1, \ldots, n$, then $\sum_{i=1}^{n} |S_i| \geq p \times n$. For each query, $(i, j)$, we have $i \in S_j$ and $j \in S_i$. So, $\sum_{i=1}^{n} |S_i| = 2 \times |B|$. If item $i$ is utilized in at least $p$ queries for each $i = 1, \ldots, n$, then $2 \times |B| \geq p \times n$. In other words, $|B| \geq \frac{p \times n}{2}$. If the number of queries utilized by the algorithm, $|B|$, is less than or equal to $\frac{p \times n}{2}$, then at least 1 item will be involved in fewer than p queries. Take $p = n - (l_1 + l_2)$. If fewer than $\frac{(n - (l_1 + l_2)) \times n}{2}$ queries are made, then at least one item will be involved in less than $n - (l_1 + l_2)$ queries. If that item belongs to $C_1$ or $C_2$, then the algorithm will not be able to determine if the item is in cluster $C_1$ or $C_2$. Thus, in the one-round case, if fewer than $\frac{(n - (l_1 + l_2)) \times n}{2}$ queries are made, it is not possible to always determine the clusters. $\square$

The previous result is consistent with Mazumdar & Saha (2017a) but our result provides a tighter lower bound and is not asymptotic. For more details, see Related Work.

**Corollary 1.** *If there are $k$ clusters, then the lower bound for the number of queries that are required to determine the clusters in one round is $\frac{(n - \frac{n}{k} - \frac{n-1}{k-1}) \times n}{2}$. If there are $k$ clusters, then the smallest cluster, $l_1$, must have at most $\frac{n}{k}$ items. Furthermore, the second smallest cluster, $l_2$, must have at most $\frac{n-1}{k-1}$ items. We can prove that $l_2 \leq \frac{n-1}{k-1}$.*

Assume for contradiction that $l_2 > \frac{n-1}{k-1}$. Then, every cluster other than the smallest cluster must contain at least $\frac{n-1}{k-1}$ items. Using this, we can obtain a bound for $n - l_1$, which is $\frac{n-1}{k-1} \times (k-1) = n - 1$. This means that the size of the smallest cluster is 0, a contradiction.

**Corollary 2.** *Utilizing the inequality $\frac{n}{k} \leq \frac{n-1}{k-1}$, we obtain from the prior corollary the lower bound of the form $\frac{(n - 2 \times \frac{n}{k}) \times n}{2} = \left(\frac{1}{2} - \frac{1}{k}\right) \times n^2$. This shows that for large $n$, this bound is asymptotically $\Omega(n^2)$, and for large $k$, the coefficient of $n^2$ is close to $\frac{1}{2}$.*

Our results for one-round algorithms provide a lower bound of the number of queries that are required to determine the clusters of a set of $n$ items into $k$ clusters when the sizes of the two smallest clusters are known, which is $\frac{(n-(l_1+l_2))\times n}{2}$ queries. Our results provide more than an order-wise asymptotic lower bound since we provide a precise lower bound for the number of queries that are required to determine the clusters of any number of items that are to be queried, a novelty.

Furthermore, it is easy to see that the lower bound is tight. Regardless of the algorithm, if fewer than $\frac{(n-(l_1+l_2))\times n}{2}$ queries are made in the first round, there will always be at least one item that has not been directly compared against at least $\ell_1 + \ell_2$ items. Thus, if that item is in either $C_1$ or $C_2$ and the items it has not been compared against are items in $C_1$ and $C_2$, it is not possible to determine if the item is in $C_1$ or $C_2$, or if it is in a single item cluster. Furthermore, as proved above, regardless of the algorithm, any time fewer than $\frac{(n-(l_1+l_2))\times n}{2}$ queries are made, there exits a clustering where an item in either the smallest cluster or the second smallest cluster has not been compared with any other item in its cluster. So, any time fewer than $\frac{(n-(l_1+l_2))\times n}{2}$ queries are made, it is not possible to determine the clusters of all the items.

Our main result for two-round algorithms is stated in the following theorem.

**Theorem 2.** *For a two-round algorithm, suppose that $m$ queries are made in the first round. Then, if the size of the largest cluster is $\ell = l_k$, then the lower bound for the number of queries that are required to determine all clusters in two rounds is $\binom{n - \frac{m}{n} \times \ell}{2}$.*

*Proof.* We consider a randomly chosen query, $X = (a, b)$, chosen from the set of possible queries $A$. Then $f(X)$ is a random variable that takes value 0 if $a$ and $b$ are not in the same cluster, or 1 if $a$ and $b$ are in the same cluster. We calculate $P(f(X) = 1)$. This is equivalent to the probability that for a fixed query, $(a, b)$, $a$ and $b$ are in the same cluster when the clustering is randomly chosen from the collection of all possible clustering for the set $S$.

$$P(f(X) = 1) = \sum_{i=1}^{k} P(f(X) = 1 | a \in C_i) \times P(a \in C_i) \tag{2}$$

$$= \sum_{i=1}^{k} P(b \in C_i | a \in C_i) \times P(a \in C_i) \tag{3}$$

$$= \sum_{i=1}^{k} \frac{|C_i|}{n} \times \frac{|C_i| - 1}{n - 1} \leq \sum_{i=1}^{k} \left(\frac{C_i}{n}\right)^2 \leq \frac{\ell}{n} \tag{4}$$

We define $d := \frac{m}{n}$. Consider the expected number of queries where $a$ and $b$ are in the same cluster for $d \times n$ queries in the first round. Let $X_1, \ldots, X_{d \times n}$ be the queries in the first round. Then the number of queries for which both elements are in the same cluster is

$$\mathbb{E}\left(\sum_{i=1}^{d \times n} f(X_i)\right) = \sum_{i=1}^{d \times n} \mathbb{E}(f(X_i)) \tag{5}$$

$$\leq (d \times n) \frac{\ell}{n} = d \times \ell \tag{6}$$

This is the expected number of queries that result in the elements of the query being in the same cluster. Recall that each time a query reflects that the elements are in the same cluster, the number of preclusters is reduced by 1, so $n - d \times \ell$ is the lower bound on the expected number of preclusters after $d \times n$ queries.

We showed that the expected number of queries where $(f(X) = 1)$ in a set of $d \times n$ queries that were made in the first round is bounded by $d \times \ell$. We can interpret this bound on the expected value as a bound on the expected number of queries for which both elements are in the same cluster in a given set of $d \times n$ queries made in the first round, where the expected value is taken with respect to all possible clusterings of the set $S$. This means that there is some clustering for which the number of queries where both elements of the query are in the same cluster is less than $d \times \ell$. Therefore, for a given set of $d \times n$ queries in the first round, there is a clustering of the set $S$ such that the number of queries for which both elements of the query belong

to the same cluster is bounded by $d \times \ell$. Recall that each time we process a query for which both items are in the same cluster the number of preclusters is reduced by at most 1. Thus, the number of preclusters for that set of queries is lower bounded by $n - d \times \ell$ after $d \times n$ queries are made in the first round.

We next show that the number of queries required to determine all clusters in two rounds is $\binom{n-d\times\ell}{2}$. In other words, if there are $(n - d \times \ell)$ preclusters after the first round, at least $\binom{n-d\times\ell}{2}$ total queries must be made in the first and second rounds.

After the first round of querying, items that are found to be in the same cluster are indistinguishable, so we can label items in the same precluster with the same integer. Let $L$ be the set of labels. There are $(n - d \times \ell)$ different labels. We will show that every query in $R = \left\{ x \in (a, b) : a \in L, b \in L, a < b \right\}$, must be made. Let $L_1$ be the set of items that are labeled $a_1$ for some $a_1 \in L$. Let $L_2$ be the set of items that are labeled $a_2$ for some $a_2 \in L, a_2 \neq a_1$. Suppose there is no query involving an item in $L_1$ and an item in $L_2$. Then, if $f(x) = 0$ for all queries $x$ between items in the set $L_1 \cup L_2$ and items in the set of remaining items, the algorithm will not be able to distinguish whether items in $L_1$ and $L_2$ are in the same cluster. $\qquad\square$

It is important to note that these $\binom{n-d\times\ell}{2}$ queries are not all made in the second round of querying. Every time $f(x) = 0$, whether in the first or second round, two preclusters are compared and determined not to be in the same cluster.

**Corollary 3.** *Suppose that all the clusters are of equal size. Then, $\ell = \frac{n}{k}$ and the number of queries is at least $\binom{n-\frac{d\times n}{k}}{2}$.*

We could also say that if the sizes of clusters are approximately equal, this lower bound holds approximately. As was the case with the one round algorithms, we note that under the constraint of $d \leq \frac{n}{\ell}$, we have developed a precise lower bound for the number of queries that are required to cluster a set of $n$ items into $k$ clusters for any $n$ and $k$, not just an order-wise lower bound that only holds for large $n$, as is the case with many analyses.

**Corollary 4.** *A lower bound for the expected number of queries for any algorithm that are required to cluster all the items is $\binom{n-d\times\ell}{2}$. If the expected number of queries for which both elements are in the same cluster less than $d \times \ell$, a lower bound for the expected number of preclusters is $(n - d \times \ell)$, and thus a lower bound for the expected number queries that are required to cluster all the items in two rounds is $\binom{n-d\times\ell}{2}$, using the fact that the expected value of a random variable squared is greater than or equal to the square of the expected value of a random variable.*

**Corollary 5.** *We can use the information above to determine a lower bound for the expected number of queries that are required to cluster the items when the sizes of the clusters are not known but it is known that $P(a \in C_i) = \frac{1}{k} \ \forall a \in S$ and $\forall i \in 1, \ldots, k$ and the cluster that a given item is in does not affect the probability that any other item is in a given cluster. In this case, the expected number of queries that are required to cluster the items is at least $\binom{n-\frac{d\times n}{k}}{2}$.*

*Proof.* Clearly, for a fixed query $x = (a, b)$, $P(f(x) = 1) = \sum_{i=1}^{k} (a \in C_i) \times P(b \in C_i) = \frac{1}{k}$ so the expected number of queries that result in the elements of the query being in the same cluster is $\mathbb{E}\left(\sum_{i=1}^{d\times n} f(x_i)\right) = \sum_{i=1}^{d\times n} \mathbb{E}(f(x_i)) = \sum_{i=1}^{d\times n} P(f(X_i) = 1) \leq (d \times n) \frac{1}{k} = \frac{d\times n}{k}$. This implies that a lower bound for the expected number of preclusters remaining after the first round of querying is $\left(n - \frac{d\times n}{k}\right)$, so a lower bound for the expected number of queries that are required to cluster all the items is $\binom{n-\frac{d\times n}{k}}{2}$. $\qquad\square$

We recognize that our results for the two round case say little about the sizes and number of preclusters that result after the first round. Information in this regard is potentially useful when the clusters are to be approximated and the final set of clusters produced by the algorithm is not certain to be the underlying clustering of the items. We use a corollary of Turan's theorem (Berge, 1985) to derive information about the number and sizes of preclusters that remain after the first round. We remark that the work of Mazumdar & Saha (2017a) also uses Turan's theorem in their proofs.

**Theorem 3.** *If $d \times n$ queries are made in the first round, in the worst case there will be a set of at least $\frac{n^2}{n+2\times(d\times\ell)}$ items that have not been found to be in the same cluster as any other item after the first round. In other words, there will be, in the worst case, at least $\frac{n^2}{n+2\times(d\times\ell)}$ single element preclusters after the first round of querying.*

*Proof.* We can reformulate the problem as a graph theoretic problem if we identify elements with nodes and queries $x$ where $f(x) = 1$ with edges. We call a subset of the nodes of a graph stable if no pair from it is connected by an arc. We can then obtain a lower bound on the size of the largest independent set in the graph, which corresponds to a lower bound of the expected number of items in the first round that have not been found to be in the same cluster as any other item. Let

$$g = \min\Big\{\max\big\{h|G \text{ contains an independent set of size } h\big\}|G \text{ is a graph with } c_i \text{ nodes and } m \text{ edges }\Big\}.$$

According to the corollary of Turan's theorem, it can be shown that $g \geq \frac{c_i^2}{2\times m+c_i}$ (Berge, 1985). In our case, $c_i = n$ as we have $n$ items to cluster, and $m = d \times \ell$, since in expectation, there are $d \times \ell$ queries between items where the two items are found to be in the same cluster, so there must exist a clustering such that there are $d \times \ell$ queries between items where the two items are found to be in the same cluster. So, in the worst case, there are at least $\frac{n^2}{n+2\times d\times\ell}$ items that have not been found to be in the same cluster as any other item after the first round of querying. $\square$

**Corollary 6.** *If $d \leq \frac{n\times k-n}{2\times\ell}$, there will always be at least one cluster where none of the items in that cluster have been compared to any other element in the cluster in the worst case. If $d \leq \frac{n\times k-n}{2\times\ell}$, $\frac{n^2}{n+2\times d\times\ell} \geq \frac{n}{k}$. So, since the size of the smallest cluster is always less than or equal to $\frac{n}{k}$, we know that it is always possible for at least the items of the smallest cluster not to have been compared with each other.*

**Theorem 4.** *If $d \times n$ queries are made in the first round, there will always be a set of at least $\frac{n^2}{2\times d\times n+n} = \frac{n}{2\times d+1}$ items that are not directly compared against each other in the worst case.*

*Proof.* The proof of this is a direct application of the corollary of Turan's theorem above, where items are identified with nodes and comparisons with edges in the obvious way. $\square$

**Corollary 7.** *If $d \times n$ queries are made in the first round and $\frac{k-1}{2} \geq d$, there will always be at least one cluster where none of the items in the cluster are compared with any of the other items in the cluster in the worst case. When the set of items where no item has not directly been compared against one another is at least of size $\frac{n}{2\times d+1}$, if $\frac{k-1}{2} \geq d$, simple algebra shows that at least $\frac{n}{k}$ items, the upper bound for the number of items in the smallest cluster, will not be directly compared against each other.*

## 5 Conclusion

Sequential querying may be impractical because of the time involved in waiting for the answers to come. A more realistic approach involves sending out a number of queries at the same time and deciding the next step based upon an analysis of all the responses received, which is called querying in rounds. In the present work, we provide lower bounds for the number of queries required to cluster a set of items in one and two rounds in the worst case and in expectation, given the number of clusters and sizes of the smallest and second smallest clusters. Additionally, we provide lower bounds for the expected number of queries when an item is in any given cluster with equal probability. We determine lower bounds for the number of single element preclusters after the first round of querying, the number of items that are not directly compared against each other after the first round of querying, and the number of preclusters after the first round of querying that have not been compared against each other. In the future, we will explore the possibility that the answers to the query are noisy. We also wish to find lower bounds for crowdsourced clustering in more than two rounds.

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
