# OpenReview forum: "On Lower Bounds for the Number of Queries in Clustering Algorithms"
_TMLR — Rejected by TMLR_

### Review · Reviewer_HTP7 · 2023-07-03

**Summary Of Contributions:**

The paper uses the probabilistic method to state finite-sample, worst-case lower bounds on the query complexity of exact clustering. This setting assumes either 1 round or 2 rounds of queries to a perfect access pairwise similarity oracle. The lower bounds depend on the largest, smallest, and 2nd smallest cluster size.

**Audience:**

No

**Broader Impact Concerns:**

No broader impact concerns

**Claims And Evidence:**

Yes

**Requested Changes:**

### Critical
- Condense the current manuscript to at most 6 pages (potentially 4 pages).
	- For example, remove either the last paragraph of Section 4 or Section 6. Similar space-saving edits can be made throughout.
- Increase the relevance/scope of contributions by considering additional aspects of the problem, for example: noisy oracle, approximation algorithms, average-case performance, >2 query rounds. Any of these directions would make the paper much more interesting to the TMLR audience.
- Fix the major typos in the statement of Theorem 2 and on line (4) of its proof.
- Update terminology "arcs" and "stable set" to "edges" and "independent set," consistent with graph theory conventions.
- There are no numerical experiments in Section 5. Either conduct experiments on clustering instances or rename the section to "Comparison of Lower Bounds".
- Clarify that Corollary 6, Theorem 4, and Corollary 7 ("there will always be...") are still worst case statements.
- This sentence is not true: "The term active refers to the fact that the answers to the queries are provided by workers who typically are members of the public".
- Cite [1] when setting up the use of Turán's Theorem to analyze query complexity in Theorem 3.
- Cite any number of seminal papers on batch active learning, e.g. [2], to make it clear that using 2 rounds of queries is not novel.

### Non-critical
- Organization: Right now the first substantive description of the method is at the end of page 3 and the first substantive description of results is in the middle of page 4 (nearly halfway through the paper). Mention important background, informal theorem statements (namely noise-free assumption and lower bounds in terms of various cluster sizes), and the main proof techniques (probabilistic method, Turán's Theorem), earlier in the paper.
- Use \citet{} instead of \citep{} to condense citations


### Minor Typos
- Sentence after Corollary 7 "for the lower bounds for the..." is not complete.
- Section 2: "theoretic lower bounds" should be "theoretical lower bounds".
- Section 3: n should be $n$, and k should be $k$.
- Proof 4: "on expectation" should be "in expectation".
- Match the index of the proof with the result it is proving, e.g. Theorem 3 and Proof 4.

Also, Acknowledgments are strongly discouraged in a double-blind submission because they are likely to violate anonymity

[2] Yuhong Guo, Dale Schuurmans. Discriminative Batch Mode Active Learning. NeurIPS 2007. https://papers.nips.cc/paper_files/paper/2007/file/ccc0aa1b81bf81e16c676ddb977c5881-Paper.pdf

**Strengths And Weaknesses:**

### Strengths
- Results appear to be correct.
- General problem area is well-motivated.

### Weaknesses
- Focuses solely on worst-case query complexity without discussing average case performance.
- Exposition is very verbose and repetitive. For example, the last paragraph of Section 4 is redundant with Section 6.
- Many crucial typos hurt the overall clarity.
- Specific results have little relevance to theorists or practitioners:
    - Theoretical results do not significantly improve over previous asymptotic bounds, and proof techniques are not novel.
    - No methods are proposed which would improve clustering performance in practical settings.
- Use of Turán's Theorem for analyzing query complexity of clustering is not novel. See Claim 4 of [1] which is discussed in Related Work.

[1] Arya Mazumdar and Barna Saha. Clustering with noisy queries. NeurIPS 2017. http://papers.nips.cc/paper/7161-clustering-with-noisy-queries.pdf

---

> ### Author Response · Authors · 2023-08-23
>
> We are grateful to the reviewer for their useful comments and suggestions. We wanted to point out that Theorem 1 of our work is consistent with the asymptotic result of [1] but our result provides a tighter lower bound and is not asymptotic. Namely, their result shows that in the case when the number of clusters, $k$, is \textit{k}$\geq 3$, stating that if the minimum cluster size is \textit{r}, then any deterministic algorithm must make $\Omega\left(\frac{n^2}{r}\right)$ queries even when query answers are not subject to error, to recover the clusters exactly. We show that any active clustering algorithm with perfect workers that determines all the clusters with one round requires more than $\frac{\left(n-\left(l_1+l_2\right)\right)\times n}{2}$ queries where $\ell_1$ and $\ell_2$ are the sizes of the smallest two clusters. We note that our main proof technique is not the use of Turan's theorem, which is simply used to prove some later results, and that our novel proof techniques are largely contained in the proofs of Theorem 1 and 2. We fixed many of the typos mentioned and removed the Acknowledgements, thank you for noting these.
>
> [1] Mazumdar, Arya, and Barna Saha. "Clustering with noisy queries." Advances in Neural Information Processing Systems 30 (2017).

---

### Review · Reviewer_g3J9 · 2023-07-07

**Summary Of Contributions:**

The paper is concerned about clustering with queries. Specifically, the setting is considered is where two points (i,j) can be given to a perfect oracle (never makes mistakes) which would return 1 if i and j are in the cluster and 0 otherwise. While previous results have consider active (sequential) queries, this paper consider querying in batches which is practically motivated by the fact that it can be easier and less time consuming. Lower bounds for the case of two queries are established.

**Audience:**

No

**Broader Impact Concerns:**

I don't have ethical concerns about this paper.

**Claims And Evidence:**

No

**Requested Changes:**

-The paper does not seem to have been proof-read.

-Writing should be significantly improved.

-Proofs should be clear and rigorous.

-I don't see the point behind having d in the Theorem statements.

**Strengths And Weaknesses:**

Strengths:
-The problem is interesting and well-motivated.

Weaknesses:

1-The writing quality and number of typos is really surprising. Here are some things to mention:

        1A-in related work: 8th paragraph talks about the some paper in the 2nd paragraph (Ashtiani et al, 2016)

        1B-Same problem for Ailon et al and Awasthi et al

	1C-If the citation format has the author names (x et al), then why say x in x et al does something

       1D-first paragraph in related can be better written

        1E-“to do provide a polynomial-time”

	1F-Ailon et al reference appears twice (it is identical)

	1G-last paragraph in related work does not cite any paper

	1H-We define l_i = |Ci | i = 1, . . . , k.  → put comma after |Ci |.



2-I find it very difficult to follow the paper and it’s not clear to me that some proofs are correct:

      2A-In Proof 1: what is B? I don’t see where it’s define. Further, if p \leq n then why would we have for each query (i,j), i \in S_j and j \in S_i. We can construct an example where this does not happen.
      2B-In Proof 2: (4) there should be n^2 before the last inequality and In don’t see why in the last inequality the square would disappear. Further, inequality (6) upper bounds the expected number of queries yet the rest of proof does not handle the fact that it is an expectation not deterministic.


3-Why is the parameter d kept in the statements of most theorems. It is not a parameter describing the instance but is decided in the query. You should choose d which minimizes the number of queries and then write the final number of queries. This manyof the theorem/corollary statements non-informative, not directly at least.

4-On page 4, observation 2 says if the number of clusters k is known and the preclusters are now k+1, then we do not need another query. I don’t see why this is correct. Suppose k=2 and  we have k+1=3 clusters, one point i is alone, we would still not know if i belongs to the first or second cluster.

5-Corollary 1: should have -\frac{n-1}{k-1} not +


6-The fact that the data exists mostly in a metric space is ignored and not utilized at all which is odd in a clustering setting



7-The paper only derives a lower bound but not upper bound, correct? The plot on page 9 should say instead our lower bound not algorithm.

---

> ### Author Response · Authors · 2023-08-23
>
> Thank you for the helpful comments. We wanted to point out that $B$ is the set of all queries that are utilized by the algorithm. We added a reference in the proof of Theorem 1.  The point that for each query (i,j), i \in S_j and j \in S_i was meant in reference to the sentence following that statement, i.e., "So, $\sum_{i=1}^{n} |S_i|$ = $2 \times |B|$." In general, for each query $(i,j)$, $i$ will always be in the set S_j and $j$ will always be in the set $S_i$. The inequality in (6) is a worst case upper bound that takes into account the fact that once the queries are chosen in the first round, since nothing is known, we are essentially randomly choosing items to assign to the queries. While it is true that the bound is given in expectation, it provides a lower bound for the worst case scenario. Regarding the n^2 in 2B, there was a typo - thank you for catching it. We added a parentheses. Additionally, we will work to exclude $d$ from the Theorem statements. Regarding Observation 2, we simply meant that if we know the number of clusters then in the worst case, even when each item belongs to its own cluster, we never need to final query because we can deduce from the number of clusters the result of the final query.

---

### Review · Reviewer_Vyhb · 2023-07-19

**Summary Of Contributions:**

This work studies the problem of learning a clustering via same-cluster queries, that is, by asking queries in the form "do x and y belong to the same cluster?". The claim is to provide (tight) lower bounds on the number of such queries for algorithms that work in one round and in two rounds (every round consists of a set of queries whose input can be a function only of answers to previous rounds). To this end, the work present a list of theorems and corollaries.


**Audience:**

Yes

**Claims And Evidence:**

No

**Requested Changes:**

None

**Strengths And Weaknesses:**

The topic is interesting. Unfortunately, the work is written in a very informal/sloppy way.

Section 1 (Introduction) discusses the problem without defining it. It just says "the process of classification of a set of items into clusters by an oracle", but there is no model for the oracle (i.e., for what the answers to the queries are), although the discussion clearly requires it.

Section 2 (Related Work) discusses related work, but without properly defining the problem(s). As a results, it gives two (apparently?) contradicting bounds of O(nk) and Omega(n^2/r). Perhaps more importantly, Section 2 does not *compare* related work to the paper's results; it is just a list of existing publications, many of which look like mostly unrelated the present submission.

Section 4 (Results) is informal. Most formal statements are unclear and/or written in a bizarre way; similarly, the proofs are informal, repetitive, and gloss over crucial points, and there are some debatable statements.

Theorem 1 gives a lower bound of (n-L)n/2 non-adaptive queries where L is the sum of the sizes of the two smallest clusters. However, this is not true for every clustering. Consider indeed the clustering consisting of k=3 clusters of sizes respectively n-2,1,1. For every element x choose other distinct 4 elements and make queries against them. If x is in the first cluster then we have at most 2 "no" answers, otherwise we have at least 3 "no" answers. Thus we can reconstruct the clustering with only 4n queries, while the lower bound is (n-2)n/2. The proof's argument indeed fails: it says that, for any two fixed clusters C1 and C2, if x belongs to C1 \cup C2 and we make fewer than n-|C1 \cup C2| queries then we won't be able to determine to which cluster among C1,C2 x belongs to. (In fact I note that the proof does not actually even *prove* that; it just literally says that "the algorithm will not be able to determine if the item is in cluster C1 or C2"). Anyway, this would hold perhaps for learning the *indices* of the clusters, but not for learning the clustering up to a relabeling, at least when |C1|=|C2|=1. It may still be the case that the theorem holds for *some* clustering where the two smallest clusters have total size L, but the proof's argument is still flawed.

Below Theorem 1 the paper claims that "it is easy to see that the lower bound is tight". This means that it is achievable by some algorithm, but the rest of the paragraph does not prove this. Instead, it repeats the proof of Theorem 1 in an informal way.
The section contains several corollaries that are written again in a weird way; statements are mixed with what look like parts of the proofs; see for instance Corollary 1.

The proof of Theorem 2 seems to prove a different bound than the one in the statement. In particular the statement claims a bound of n - d * (\ell choose 2), while the proof ends up with (n - d * \ell choose 2). I am not sure whether this is a typo; it is repeated several times throughout the proof, and Corollaries 3 and 4 also claim bounds akin to the second one. The proof of Theorem 2 is again informal and repetitive.

Section 5 (Numerical Experiments) consists of a single sentence and a plot. The sentence says "We compare our lower bounds for the one-round case to those of Mazumdar & Saha (2017a) where k = 20 and the clusters are of equal size in Figure 5" and the plot shows two curves, "Our Algorithm" and "Algorithm in [4]". I am not sure what algorithms are these; I think the authors meant "lower bound". In any case I do not think these are "numerical experiments"; these are just expressions evaluated for certain values of n and k.

I remark that no result in this work is proven to be tight (in any sense).

---

> ### Author Response · Authors · 2023-08-23
>
> We thank the reviewer for the useful feedback. Regarding the comments on Section 1 (Introduction) and Section 2 (Related Work), we moved the Problem Setup section to now be before the Related Work so the exact problem formulation could be compared to the related works. Regarding tightness of Theorem 1, our argument for tightness was meant to show what would happen in a worst case scenario if there were no direct comparison between items from the smallest two cluster, which is that there would be no way of knowing whether or not those two preclusters should be in the same cluster, so an additional query would be needed to exactly recover the clusters. While the tightness remark may appear similar to the proof of Theorem 1, it is a worst case result, so one might expect the result to be somewhat based on a counterexample. There was a typo in Theorem 2 statement, thank you for catching it. We agree that the Numerical Experiments section is not entirely necessary and have removed it.

---

> > ### Comment · Reviewer_Vyhb · 2023-08-23
> >
> > I thank the authors for their effort. However, neither the reply nor the new revision seem to address my concerns. For instance:
> >
> > - the proof of Theorem 1 has the same flaw that I raised (it basically ignores the most crucial point, i.e., proving that one "cannot" find out the cluster of a certain element).
> >
> > - the "tightness" argument is still there (and again: that argument does not prove any tightness, it just repeats the theorem's claim in more informal terms).
> >
> > - all statements and proofs are still written in an informal and mathematically poor way.

---

> > > ### Author Response · Authors · 2023-08-23
> > >
> > > I thank the reviewer for the feedback. Regarding the comment on the proof of Theorem 1, it seems that any time at least one cluster is of size 1, all the queries (n(n-1)/2) need to be made, which is consistent with the results in Theorem 1. The reason for this is that if a single query is not made and that query happens to involve the item in the cluster of size 1, then it is impossible to determine whether that item is in its own cluster or if that item is in the cluster of the other item in the query that is not made. Thus, I think that the lower bound is consistent in the case of the counter example. Also, regarding the tightness argument, when I meant to say is that suppose that an item has not been compared to at least $\ell_1+\ell_2$ items. Then, if that item is in either $C_1$ or $C_2$ and the items it has not been compared against are items in $C_1$ and $C_2,$ it is not possible to determine if the item is in $C_1$ or $C_2$, or if it is in a single item cluster. I added a clarification in the manuscript, thank you for the comment.

---

### Decision · Action_Editors · 2023-09-08

**Recommendation:** Reject

**Comment:**

Thank you for your submission to TMLR. We had three expert reviewers evaluate the submission. One of the reviewers opines the absence of interest from the community and the other two reviewers say that the claims are not supported by clear and accurate evidence. The proofs should be written in a more rigorous manner to enable the reviewers to verify their correctness.

**Audience:**

One reviewer opines "no" and the other two opine "yes". At this point, this is slightly subjective, but given that there is insufficient evidence to justify the claim, the paper cannot be accepted due to that criterion anyways.

**Claims And Evidence:**

The formal parts of the paper are also written quite informally, due to which reviewers could not see accurate, convincing and clear evidence supporting the paper's claims.